# Cooperative environmental engineering via biofilm formation can stabilize consumer-resource systems

Bryan K. Lynn[1¤*], Patrick De Leenheer[1,2], Benjamin D. Dalziel[1]

1 Integrative Biology, Oregon State University, Corvallis, Oregon, United States of America,
2 Mathematics, Oregon State University, Corvallis, Oregon, United States of America

¤ Current address: Ecology and Evolutionary Biology, University of California Irvine, Irvine, California, United States of America
* bklynn@uci.edu

## Abstract

Cooperation can stabilize consumer-resource dynamics, preventing over-exploitation driven by individual self-interest. The maintenance of cooperation in such systems is often attributed to individual-level behaviors, such as punishment of defectors, however, an alternate and under-explored path to stability involves environmental engineering by cooperative consumers, who may modify the environment to favor cooperators. Microbial biofilms are an important instance of cooperative resource use involving environmental modification. Here, we demonstrate that biofilms can stabilize cooperative populations against cheating by giving preferential access of resources to cooperators, ensuring positive growth rates for cooperators when rare. We show that collective environmental modification offers pathways to stability across a broad parameter space, encompassing a range of rates for physiological processes, social behaviors, and environmental interactions. Including cooperative environmental modification in models of consumer-resource dynamics opens novel directions for understanding and managing ecological and evolutionary dynamics in social systems.

## Introduction

Cooperation can play an important role in the maintenance of shared resources, as seen in global fisheries or communal land use [1–10]. In population biology, cooperation is defined as a behavior that benefits other individuals and should therefore reduce the relative fitness of the actor, all else equal [11–13]. However, cooperation is prevalent across a wide range of species and ecological contexts, raising the question of what mechanisms promote its evolution and maintenance (i.e., the cooperation problem) [12–17].

**Data availability statement:** The code and the data it produced that support the findings in this study are openly available in figshare at https://doi.org/10.6084/m9.figshare.c.7672832.

**Funding:** The author(s) received no specific funding for this work.

**Competing interests:** The authors have declared that no competing interests exist.

A common type of cooperation in biological populations involves the production and use of public goods, such as resource-gathering secretions in bacteria or alarm calling in animals [12–14,18–20]. Public goods can be used by non-producers (also called "social cheaters") who do not pay the cost of producing them. This gives a selective advantage to cheaters and risks over-exploitation of resources. If that resource is required for survival, the population will then collapse—illustrating the Tragedy of the Commons [4,13].

Previous theory on the cooperation problem has identified many mechanisms that maintain cooperation and avoid a Tragedy of the Commons. Examples include kin selection, reciprocity, punishment, and spatial structure [13,14,17,21]. However, the role of ecosystem or environmental engineering, the non-trophic changes made by an organism to their environment [22,23], has been under-explored as a possible solution to the cooperation problem despite its prevalence across nature appearing in numerous ecological systems. For example, bees cooperate to produce propolis, a resin-like material made from tree sap, beeswax, and their own discharges, and use it to reinforce their hive. This results in regulation of humidity and temperature, and helps ward against unwanted bacteria, viruses, fungi, and pests that would exploit their resources such as hive beetles. Microbes are also highly capable of changing their environments. Cyanobacteria catalyzed the Great Oxidation Event where the methane in Earth's atmosphere was replaced by oxygen transforming the planet and all life on it, while also suffocating many of the existing anaerobic species present [24–27].

Environmental engineering is a concept closely related to that of niche construction. While both concepts aim to correct the bias of traditional theories that treat environments as given and exogenous, Niche Construction Theory differs from environmental engineering in that it is focused on how such environmental modifications affect selection pressures and evolutionary responses [22,23]. Here, we use the language of environmental engineering instead of niche construction because we do not directly explore possible selection pressures and subsequent adaptations—a necessary component of Niche Construction Theory—and instead focus on the ecological process of niche construction (i.e., environmental engineering).

Investigations that have explored the role of environmental feedbacks in cooperative populations have primarily relied on the use of phenomenological modeling [28–34]. This is unsurprising given foundational investigations into the cooperation problem also used phenomenological game-theory models [35]. However, while these phenomenological models are useful in predicting what strategies might evolve, they do not consider the underlying biological mechanisms and ecological dynamics resulting in calls towards a more mechanistic understanding of how cooperation may evolve and be maintained in ecosystems [17,36–38]. The role of ecological feedbacks in particular can be quantified by modifying consumer-resource models [39,40] to include cooperation [41], and classic resource-competition models, such as those that underpin niche theory, often assume that species draw from an exogenously-driven resource pool [39,42–45]. Although mechanistic in nature, such models have rarely considered how organisms may modify their environments

in ways that affect resource availability—for example, by increasing resource supply rather than solely depleting it via consumption.

Populations of social microbes make good models for studying cooperative consumer-resource dynamics, showing a range of dynamics including fixation of either cooperation or cheating [46–49], stable coexistence of both cooperators and cheaters [46,48], and oscillatory patterns [50,51]. These varying dynamics may be driven by the different mechanisms used to sustain cooperation in microbes including reciprocity [52], punishment [53–55], privatization of public goods [56–59], and spatial structure [21,60–64].

Many microbes engineer their environment through the formation of biofilms, which are clusters of cells embedded in a self-produced matrix. Producing a biofilm requires cooperation through the coordinated production of the extracellular matrix, which typically occurs when populations are sufficiently dense, which is determined by quorum sensing. As with propolis production, biofilms modify access to resources, such as by slowing diffusion of nutrients outward [65]. Found on the slime-covered rocks in a pond to the plaque formed on teeth, biofilms are common across ecosystems and have been shown to be a key mediator of population survival [66,67]; for example, similar to how the bee's propolis buffers against outside effects, biofilms provide a refuge for microbial populations. This is especially relevant in health care, as chronic infections occur when the biofilms are able to mitigate the effects of treatment and the host's immune system [68].

While physical and molecular aspects of biofilms have been extensively studied, their role in ecological and evolutionary dynamics is less explored. Taken together with the parallels between microbial biofilms and other systems of cooperative environmental engineering, biofilms are an ideal system for investigating the role of cooperative environmental engineering on population dynamics.

Microbes also cooperate by producing various extracellular products into the environment that are shared by the entire population (i.e., public goods)—some of which can increase the amount of nutrients available for consumption. Yet, in many natural systems, shared resources are not perfectly public. Environmental and social structures can lead to partial privatization, where public goods are preferentially accessed by producers. Crucially, the extracellular polymeric substances that make up a biofilm reduces diffusion of these public goods making them more readily retained by the cells that produced them [65,69–71], and effectively partially privatizing public goods within the biofilm [65,72]. Such mechanisms of privatization are a general solution to prevent population collapse, or a Tragedy of the Commons [73]. In our system, this preferential access to the public good by cooperative cells within the biofilm can increase the relative fitness of cooperation, and subsequently, result in a stabilizing effect on cooperative microbial populations in the presence of social cheaters [58,65].

Both biofilm formation and public good production are common microbial behaviors, for example, the social opportunistic pathogen *Pseudomonas aeruginosa* produces a protease enzyme that degrades larger protein molecules into smaller consumable amino acid products and produces extracellular polymeric substances to form biofilms making it resistant to antibiotics. The production of these extracellular enzymes or biofilm polymeric substances, in addition to many other genes, is controlled by quorum sensing which regulates gene expression in response to cell density [74–76]. Thus, the linking of these two systems was not arbitrary, as quorum-sensing regulated protease secretion and biofilm matrix production is common in microbial species like *P. aeruginosa* and *Vibrio cholerae* [14,76–79]. The chemostat is a good environment for exploring social microbial populations, because chemostat models provide a tractable framework with well-defined parameters allowing for precise, mechanistic modeling of ecological and evolutionary dynamics.

In our system (Fig 1A), a cooperative microbe uses quorum sensing to regulate production of two related products: (1) an extracellular protease, which breaks down environmental proteins into usable nutrients and (2) a biofilm matrix on the edge of the chemostat vessel. The presence of the biofilm results in two distinct environments which the cooperative

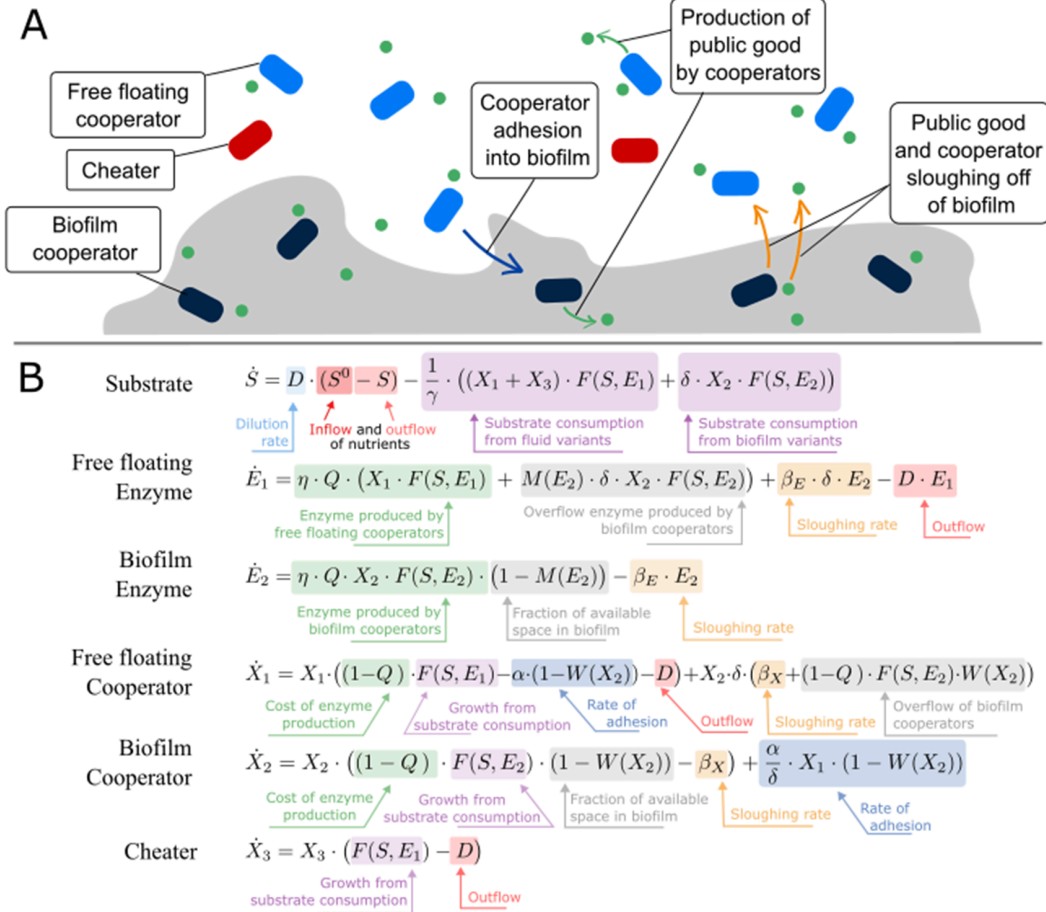

**Fig 1**. **Chemostat model.** A: Cartoon illustrating the exchange between the biofilm and fluid medium of the chemostat. B: Chemostat model describing the change in concentrations of nutrient substrate ($S$), public good enzyme produced by the cooperators in the fluid ($E_1$), public good enzyme produced by the cooperators in the biofilm ($E_2$), the free floating enzyme-producing bacterial cooperator in the fluid medium ($X_1$), the enzyme-producing bacterial cooperator in the biofilm ($X_2$), and the free floating bacterial cheater in the fluid medium which does not produce any proteases ($X_3$).

quorum sensing microbe can occupy: free floating in the liquid medium within the chemostat or embedded within the biofilm formed along the edge of the vessel. A pivotal aspect of this system is that the cheater does not produce enzymes nor the biofilm matrix. This is because the production of both is regulated by quorum sensing, a gene commonly lost in microbes due to its high metabolic cost [78,80]. Although the social cheater does not pay the fitness cost of quorum sensing, they do require the presence of enzymes in the environment for survival. Thus, cheaters may out compete and exclude cooperators, leading to the collapse of both populations as enzyme production ceases.

Here, we model the potential for biofilm formation to stabilize these cooperative consumer-resource dynamics in a mathematical model of a chemostat (Fig 1B), and examine how varying parameters governing physical conditions, vital rates, physiological constraints, and social factors influence the potential for stable coexistence—defined as the long-term persistence of species initially present—in scenarios with and without cheaters, and with biofilm matrix production either enabled or disabled. The results indicate that by structuring the availability of public goods to benefit cooperative consumers, environmental modification via biofilm formation can play a key role in the stability of cooperative consumer-resource dynamics.

## Results

### Biofilm formation allows for stable coexistence

We found that biofilm formation can exclude cheaters, or allow stable coexistence of cooperators and cheaters, for a broad range of parameter values (Figs 2 and 3). Conversely, without biofilms, cheaters cause system collapse, as expected given our system provides a representation of the cooperation problem in a microbial biological context. In Monte Carlo simulations, the frequency of stable coexistence varied across each of four growth conditions: free floating cooperators alone (cooperators that cannot form biofilm), free floating and biofilm cooperators together, free floating cooperators with cheaters, and free floating and biofilm cooperators with cheaters (S1 Fig and S3 Table). Free floating cooperators can maintain a positive stable state in the absence of cheaters (Fig 2: top left) but the addition of cheaters causes a population collapse (Fig 2: top right); however, the addition of biofilm formation can maintain a positive stable in the absence of cheaters (Fig 2: bottom left) and in the presence of cheaters (Fig 2: bottom right).

Coexistence outcomes depended on interactions among parameters. For example, a high sloughing rate of cooperators off of the biofilm ($\beta_X$) increases the likelihood of system collapse, however, increasing the adhesion rate ($\alpha$) counteracts the effect of a high sloughing rate (Fig 3A). Similarly, for influent nutrient concentrations above a minimum threshold, stability required that the cost of quorum sensing fall within a bounded range that decreases in response

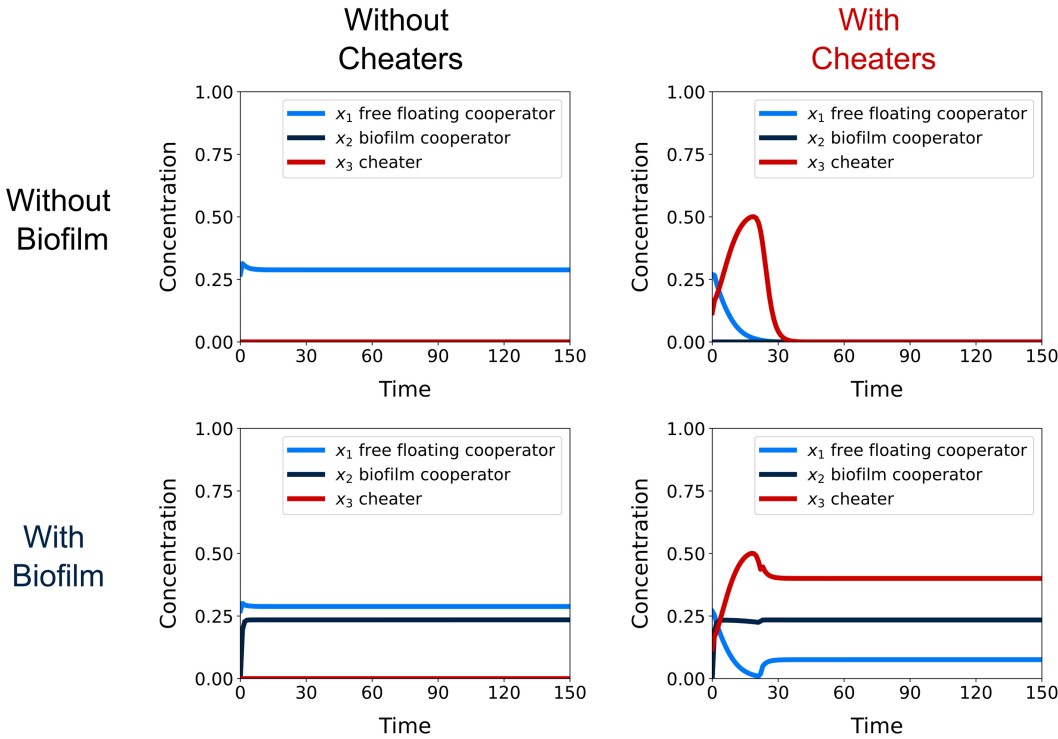

**Fig 2. Population dynamics in the presence and absence of biofilm and cheaters.** Time series showing the concentration of free floating cooperators, biofilm cooperators, and cheaters. The initial initial conditions are set to the median values of full stable coexistence (free floating cooperators, biofilm cooperators, and cheaters are all present) (S1 Fig) with the exception of two values: the adhesion rate ($\alpha$) (which determines the presence of biofilm cooperators) and the presence of cheaters ($X_3(0)$). These were either set to their median value or 0 as follows: without cheaters and without biofilm $\alpha = 0$ and $X_3(0) = 0$ (top left); with cheaters and without biofilm adhesion $\alpha = 0$ and $X_3(0) = 0.11655$ (top right); without cheaters and with biofilm adhesion $\alpha = 0.11401$ and $X_3(0) = 0$ (bottom left), with cheaters and with biofilm $\alpha = 0.11401$ and $X_3(0) = 0.11655$ (bottom right). The remaining median parameter and initial condition values were as follows: $S^0 = 0.76202, \delta = 0.08377, \eta = 0.80704, \gamma = 0.71445, \hat{E}_2 = 0.19065, \hat{X}_2 = 0.23565, \beta_E = 0.02042, \beta_X = 0.00699, D = 0.41712, Q = 0.47026, \mu = 68.07673, S(0) = 0.14862, E_1(0) = 0.13806, X_1(0) = 0.27003, X_2(0) = 0$. The addition of biofilm formation can avoid population collapse in the presence of social cheaters.

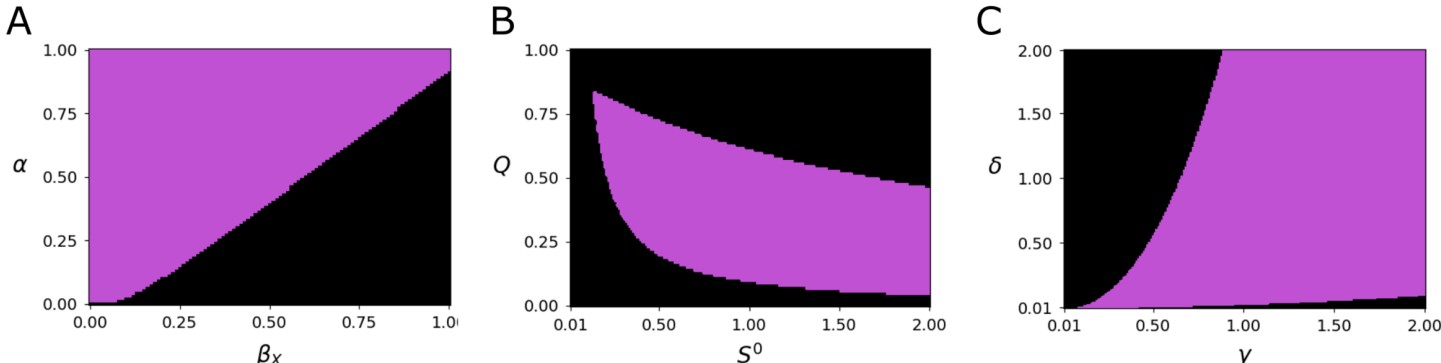

**Fig 3. Stable parameter space.** A–C: Full system stability across two parameter ranges. Purple indicates regions where full stability occurred. All other parameters and initial conditions are at their median values as described in Fig 2. Ranges for each parameter is the same range of values the parameters were randomly selected from. Four parameter pairs are shown: (A) the adhesion rate ($\alpha$) with the bacterial sloughing rate ($\beta_X$), (B) the cost of quorum sensing ($Q$) with the inflowing nutrient concentration ($S^0$), and (C) the biofilm area to fluid volume conversion ($\delta$) with the biofilm to nutrient uptake conversion ($\gamma$).

to increasing influent nutrient concentrations (Fig 3B). This is likely because the cost of quorum sensing is a fraction of the metabolic energy devoted towards enzyme production and away from growth and reproduction. As nutrient availability increases, less enzyme is needed to produce sufficient amino acids, freeing more energy for growth. We also observed tradeoffs between parameters whose relationship was more indirect. For example, increasing the biofilm to nutrient uptake conversion ($\gamma$) allows for the biofilm area to fluid volume conversion ($\delta$) to also increase while maintaining a full stable coexistence (Fig 3C). The multiplicitous parameter tradeoffs that maintain positive coexistence create a complicated stable parameter space. The shape of the stable regions in Fig 3 suggest that there is a bounded subset within which full stable coexistence always occurs; however, as parameter values extend beyond those bounds, the outcomes become dependent on the values of other parameters.

We found that the maximum microbial growth rate ($\mu$) functions as a bifurcation parameter, determining three main system outcomes: (i) washout, where neither cooperators nor cheaters persist; (ii) cooperator-only equilibrium, where cheaters are excluded and cooperators remain stable; and (iii) stable coexistence, where both cooperators and cheaters persist at steady state (Fig 4). Stable cycles appear at the transition between cooperator-only stability and a fully stable system, however, the precise location and amplitude of these cycles is dependent on other parameter values. Analyzing the Eigenvalues of the system, it was determined that a Hopf bifurcation is responsible for the stable limit cycles observed (S2 Fig).

## Isolating the social effect of biofilm formation

Stable coexistence of both free-floating and biofilm cooperators, in the absence of cheaters, establishes the minimum requirements to avoid a population collapse caused by insufficient growth or nutrient efficiency. To isolate the effects of cheating, we restricted the parameter space such that any randomly chosen set produced a stable state with positive cooperator concentrations for both types. We then added cheaters into the system resulting in three primary outcomes: full coexistence, Tragedy of the Commons, and cheater exclusion. We observed a fully stable system at a wide range of parameter values with 74.2% of the simulations being stable, 13.1% resulting in an exclusion of cheaters and 0.056% (56 simulations) resulting in a Tragedy of the Commons (S2 Fig). The remaining 12% of the simulations included idiosyncratic cases, such as where the biofilm is stable and the fluid habitat is flushed out. We include them to illustrate that the presence of biofilms stabilizing the cooperative population occurs frequently across a range of parameter values and can drastically decrease the likelihood of a population collapse.

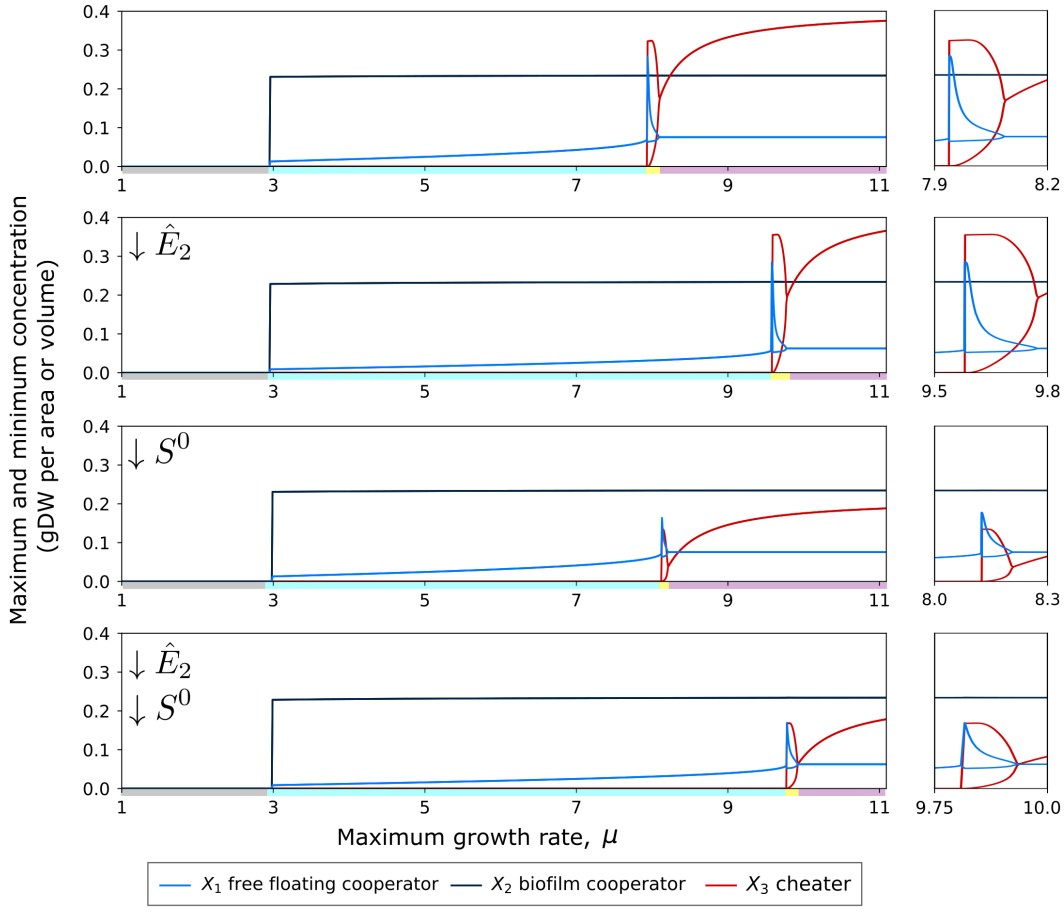

**Fig 4. Emergence of stable cycles.** The minimum and maximum concentrations of the free floating cooperator (light blue), biofilm cooperator (dark blue), and cheater (red) across two ranges of the maximum growth rate values ($\mu$): from 1 to 11 (left) and a 0.3 range around the stable limit cycle (right). Four different population conditions occur indicated by a color along the x axis: no stability (grey), cheater exclusion (light blue), and stable limit cycles (yellow), and full stable coexistence without cycles (purple). All parameters other than the maximum growth rate ($\mu$) are at their median values (Fig 2), except for two: (i) the maximum enzyme retained in the biofilm ($\hat{E}_2$) was reduced from 0.19 to 0.13 and (ii) the nutrient concentration flowing into the chemostat ($S^0$) was reduced from 0.762 to 0.5 as indicated on the top left corner of each graph. The simulations ran for 10,000 time steps, and the concentration values shown indicate the maximum and minimum concentration from the final 2,500 time steps. A stable limit cycle occurs as the system transitions from cheater exclusion to full stable coexistence (top), however, its precise location and amplitude is dependent on other parameter values.

The Tragedy of the Commons occurs under conditions that do not allow a sufficient size biofilm to grow. Thus, we see a population collapse when the following conditions are true: (i) the amount of bacterial adhesion is not sufficiently high (low adhesion rate, $\alpha$; high area to volume conversion, $\delta$) to compensate for a high cooperator sloughing rate ($\beta_X$), (ii) when the growth rate is lower (high $K_S$ and low $\mu$) and less capable of keeping up with the sloughing rate, and (iii) when there is low initial concentration of free floating cooperators paired with a higher initial concentration of cheaters (Fig 5; purple).

The exclusion of the social cheaters occurs when there is positive recruitment in the biofilm but not in the fluid environment, in which case the free-floating cooperator population is rescued due to sloughing off of the biofilm. These conditions for cheater exclusion are primarily driven by resource availability and include: (i) when there is a low influent nutrient concentration ($S^0$) and (ii) there is a higher nutrient uptake by the bacteria due to a low nutrient to biomass conversion ($\gamma$) and a high area to volume concentration conversion ($\delta$) (Fig 5; gold).

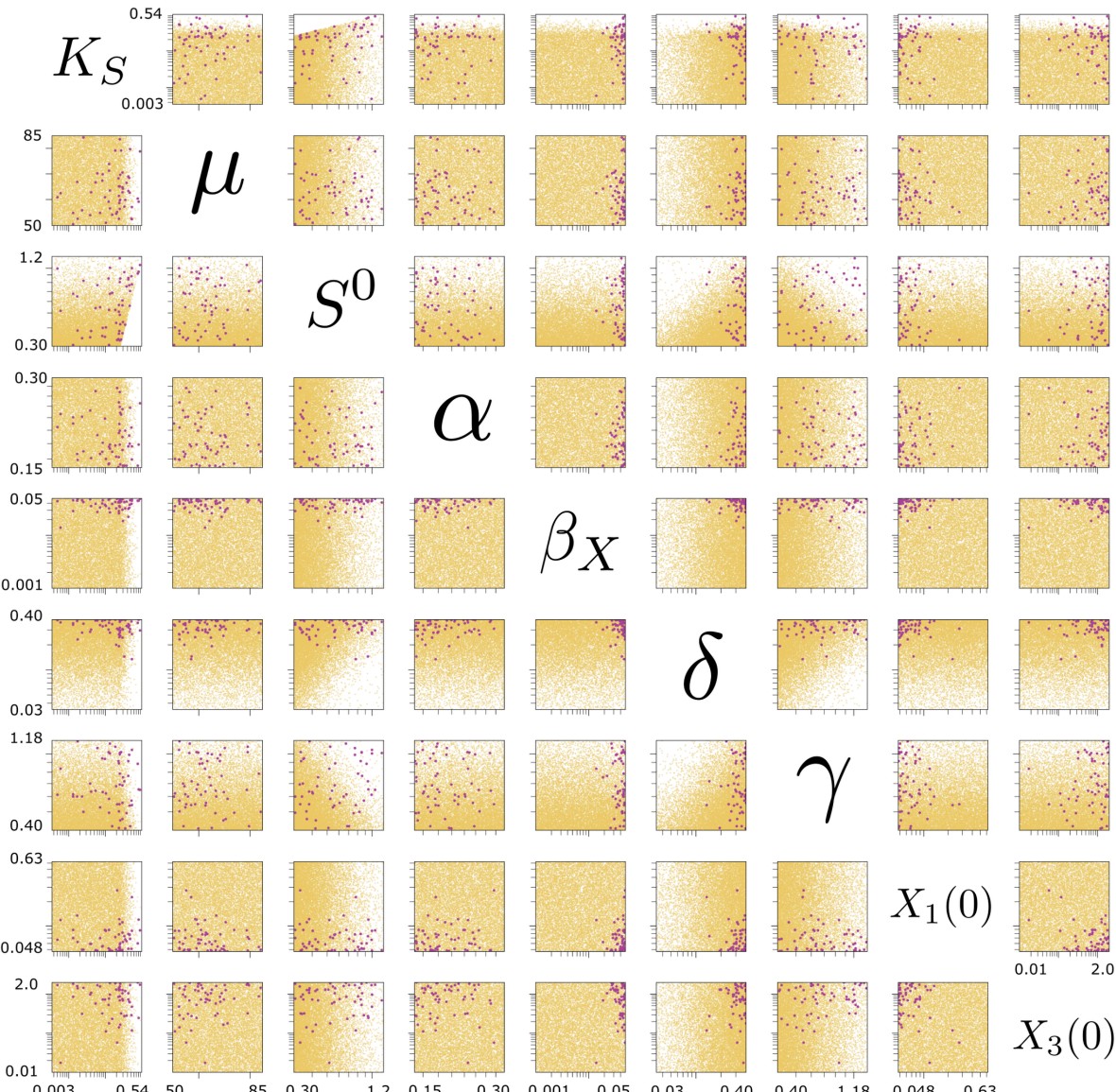

**Fig 5**. **Sensitivity analysis to deviations from full stable coexistence.** Each point represents a simulation that resulted in either a Tragedy of the Commons (purple) or exclusion of cheaters (gold). All parameter values are set to their median stable values (Fig 2) except for the pairwise perturbations indicated by parameter labels across the diagonal. Graph labels indicated across the diagonal serve as a Y-axis label to graphs to the left and right of the label and as an X-axis label to graphs above and below the label include: nutrient half-saturation constant ($K_S$), maximum growth rate ($\mu$), inflow nutrient concentration ($S^0$), adhesion rate ($\alpha$), sloughing rate of biofilm cooperators ($beta_X$), area to volume conversion ($\delta$), nutrient efficiency ($\gamma$), initial concentration of free floating cooperators ($X_1(0)$) and cheaters ($X_3(0)$). A scatter plot showing all parameter pairs as well as the simulations showing full stability can be found in the Supplemental Information, S3 Fig and S4 Fig respectively.

## Discussion

Taken together, our results indicate that biofilm formation can have a profound effect on the ecology and evolution of social cooperation for microbes in a chemostat environment. Previous attempts to find stabilizing mechanisms in a chemostat, such as through positive cooperator mutations or via punishment of the cheaters, has only resulted in the exclusion of cheaters [81,82]. This underscores the importance of expanding beyond individual level interactions, such as

punishment and reciprocity, and additionally considering the implications, impacts, and outcomes of collective behavior, such as environmental modification, into our understanding of how cooperative populations are maintained and evolve.

There was no low-dimensional set of parameters that could predict stable coexistence in the presence of cheaters (S1 Fig). This is likely because relationships between the various parameters allow for multiple routes to stable coexistence (Fig 3). This makes precise requirements for full stable coexistence non-generic and challenging to quantify, further, Hopf bifurcations occurred near the boundary of the stable parameter space creating limit cycles (Fig 4 and S2 Fig. The existence of limit cycles in nature may then be indicative of a species being on the verge of both becoming either stable or extinct. It is also worth noting that the emergence of these bifurcations in systems of differential equations used to model population interactions is not unique, for example, in Lotka-Volterra predator prey models, a Hopf bifurcation is the driving the force behind the Paradox of Enrichment in which increasing food for the prey species destabilizes the predator species.

In the absence of cheaters, certain physiological requirements need to be met for full stable coexistence. For instance, growth must be sufficiently high relative to dilution or sloughing. We found a range of parameter values in which those requirements are met and added social cheaters to isolate the social effects of biofilm formation in the presence of cheating (Fig 5). In this case, only under rare and specific conditions that hindered biofilm growth did a Tragedy of the Commons occur, demonstrating the importance of biofilm formation on the maintenance of cooperation. Further, from a resilience theory perspective [83,84], these biofilm-driven feedbacks illustrate how biofilms increase the system's capacity to absorb disturbances and subsequently delay or avoid a population collapse. These results are consistent with our understanding that one key role biofilm formation plays is stabilizing ecosystems; for example, biofilms have been shown to stabilize river sediments and serve as holdfasts for riverweeds near waterfall impact points [69,85,86]. Greater integration of biofilms into eco-evolutionary studies may reveal even more ways in which biofilms contribute to ecosystem resilience and stabilize populations.

More broadly, microbial biofilms provide a tractable model for studying the same mechanisms that underpin resilience and stability in Social-Ecological Systems [7,87]. Ostrom's Social-Ecological System Framework (SESF) [7], although typically used to describe the management of environmental resources by human communities, maps surprisingly well onto microbial public goods dilemmas. In our context, the chemostat functions as the resource system, secreted public goods act as resource units, collective action via biofilm formation plays the role of governance, and cooperators and cheaters represent the actors. According to the SESF, the resource system, resource units, governance, and actors all feedback to and interact with each other and the Focal Situation comprising of interactions and outcomes of the Focal Situation. In the case of microbes, the presence or absence of public good production are our interactions of interest and the outcomes are population stability or collapse. While cooperative microbial populations lack the traditional governance systems of human-environment social-ecological systems, their genetics and environment act as internal governance mechanisms. As such, applying the social-ecological framework to cooperative microbial populations could provide fresh insight into how collective action and other forms of internal governance sustain shared resources—processes that remain relatively understudied in the social-ecological system literature [87].

In our model we described the biofilm to be made up homogenously of cooperative cells that contribute to the production of the enzyme and biofilm matrix. In reality, biofilms are highly complex systems that can be heterogeneous in structure and composition and, as with planktonic populations, may be susceptible to invasion by non-producing social cheaters [72,79,86,88–90], however, the physiological and spatial heterogeneity is typically driven by the segregated sub-populations that form within the biofilm [79,88,90,91]. In the case of non-producing cheaters emerging in a cooperative biofilm, it has been shown that strong cooperator-only patches exist among weaker patches containing the non-producing cheaters [72]. Since more of the population within the cooperator-only patch is producing the biofilm matrix, the cooperator-only patches are thicker, denser, and therefore capable of retaining more of the public good [65,92]. A patchy biofilm has distinct zones of cooperator-only populations interacting with mixed-population zones, just like our chemostat

model. Thus, the similarities between a patchy biofilm and our chemostat model suggests similar outcomes are likely to be observed given fitness-density covariance is a stabilizing mechanism [43].

Quorum sensing has been shown to regulate hundreds of genes—including initiating gene expression for the production of public goods, like metabolic enzymes, alongside the production of the extracellular polymeric substances that make up the biofilm matrix [74–76,93]. Essentially, quorum sensing ensures that the production of the costly public good, that could otherwise make the population susceptible to social cheating, is paired with the formation of biofilms to stabilize the population should cheaters arise. By coupling the genes responsible for producing the public good enzyme with the stabilizing biofilm mechanism, cooperative populations increase their chances at avoiding a population collapse.

## Materials and methods

### Experimental design

Chemostats, continuous culturing devices for growing bacteria, are powerful tools for investigating questions surrounding the underlying mechanisms driving ecological population dynamics. Chemostat models are mechanistic by design, incorporating the population's growth, nutrient uptake rates, and production of extracellular products and public goods—all of which can be empirically measured by microbiologists. Empirical use of chemostats also allows for experimental replication of the system revealing the extent of which the dynamics are driven by deterministic or stochastic influences. Together, this allows for both an empirical and theoretical understanding of the underlying mechanisms at the root of observed shifts in population dynamics. Additionally, in a chemostat, there is a constant flow of nutrients into the culture and an outflow of the mixed microbial culture. This flow not only emulates the natural turnover of nutrients and growth/death processes seen in larger ecosystems, but it also allows for a population collapse to be observable. The validity of chemostat theory may therefore be achievable without having to infer the Tragedy of the Commons from relative fitness measurements. In general, integrating empirical and theoretical analysis of systems in which a true Tragedy of the Commons occurs with a population collapse will contribute to a more generalized understanding of the underlying mechanisms that drive the various outcomes observed in the study of cooperation and the Tragedy of the Commons.

### Model construction

We consider a chemostat model containing nutrient substrate ($S$), free floating cooperators ($X_1$), cooperators in a biofilm ($X_2$), free floating public good enzyme ($E_1$), and enzyme in the biofilm ($E_2$). We separated the respective enzyme concentrations because it has been shown that biofilms can limit diffusion of public goods, thereby retaining them within the biofilm environment [65,69–71]. The flow of the chemostat is controlled by a positive dilution rate $D$ which results in a positive inflow of nutrients at concentration $S^0$ and negative outflow of nutrients, free floating cooperators, and free floating enzyme. The cooperators distribute their energy between two functions: (i) quorum sensing behaviors such as biofilm and enzyme production and (ii) growth. The constant $Q \in [0, 1)$ is the fraction of metabolic energy allocated towards quorum sensing behaviors; this energy is converted to enzyme production with efficiency $\eta$. The remaining fraction of energy ($1-Q$) is allocated towards growth and reproduction. Since the cheaters do not contribute to the public good, it allocates the entirety of its energy towards growth.

Cooperators move from the fluid phase to the biofilm at adhesion rate $\alpha$, and both the bacteria and enzyme are sloughed off the biofilm and into the fluid phase at rates $\beta_X$ and $\beta_E$, respectively. Since the biomass per surface area is likely to be different than the biomass per volume of fluid, we use $\delta$ to convert the concentration per biofilm area to fluid phase volume. Further, different sized chemostat vessels will change the amount of surface area present and therefore also impact the amount of biofilm that can form. To account for this we let $\hat{E}_2$ and $\hat{X}_2$ be the maximum amount of enzyme and bacteria that can be in the biofilm and define functions $M(E_2) = E_2/\hat{E}_2$ and $W(X_2) = X_2/\hat{X}_2$ to be the fraction of potential biofilm space that is occupied by the enzyme and bacteria, respectively. If excess enzyme production or bacterial growth occurs in the biofilm that it cannot contain, then it will leak out into the fluid environment.

 

The per capita uptake rate function $F(S, E_i)$, where $E_i$ is either $E_1$ or $E_2$, defines the growth of the bacteria based on the concentrations of nutrient substrate and enzyme present. The per capita uptake rates of cooperators and cheaters are assumed to be the same and are given by $F(S, E_i)/\gamma$ where $\gamma$ is the yield constant in the conversion of nutrient to new biomass. This is then captured by the annotated model in Fig 1B.

We assume the function $F(S, E_i)$ is non-negative and twice continuously differentiable and satisfies the following assumptions:

**N1**

$$F(0, E_i) = F(S, 0) = 0,$$

$$F(S, E_i) > 0 \text{ when } S > 0 \text{ and } E_i > 0,$$

$$\frac{\partial F}{\partial S} > 0 \text{ and } \frac{\partial F}{\partial E_i} > 0 \text{ when } S > 0 \text{ and } E_i > 0.$$

These assumptions imply there is no nutrient uptake when nutrient or public good are absent, that there is nutrient uptake when both are present, and that with increased levels of nutrient or public good there is also an increase in uptake rates. Typical examples satisfying 1 include functions of the form $F(S, E_i) = F_1(S) \cdot F_2(E_i)$, where $F_1(S)$ and $F_2(E_i)$ are Monod or linear functions (i.e. $m \cdot S/(a + S)$ or $a \cdot S$ where $m > 0$ and $a > 0$ are parameters), the former of which are commonly used to model microbial growth [64,94,95]. For the simulations in this study, the Monod growth function $F(S, E_i) = (\mu \cdot S \cdot E_i)/(K_S + S)$ was used adding two more parameters to the model: the maximum growth rate $\mu$ and half-saturation constant $K_S$.

This model is well-posed, meaning for all non-negative initial conditions, solutions will remain non-negative and bounded. This is important for two reasons. First, it is biologically relevant, as the chemostat cannot hold infinite amounts of bacteria or other substances. Second, it has implications in numerical analysis, as solutions are less likely to be unstable or prone to error magnification (See S1 File for proof).

For reference, a list of the model parameters and variables are summarized in S1 Table.

### *In situ* details

For the simulations, we examined four different growth conditions for each combination of either presence or absence of biofilm cooperators and free floating cheaters in a system that contains free floating cooperators. The presence of biofilm cooperators was determined by a positive adhesion rate ($\alpha$); when the adhesion rate is set to 0 the biofilm does not form. The presence of cheaters is determined by a positive initial condition of cheaters; when the initial condition is set to 0, no cheaters are able to enter the system. The four growth conditions are as follows: (i) free floating cooperators are present but biofilm cooperators and cheaters are absent ($X_1 > 0$, $\alpha = X_3 = 0$), (ii) free floating cooperators and biofilm cooperators are present, but free floating cheaters are absent ($X_1, \alpha > 0$, $X_3 = 0$), (iii) free floating cooperators, biofilm cooperators, and cheaters are present ($X_1, \alpha, X_3 > 0$), and (iv) free floating cooperators and cheaters are present but biofilm cooperators are absent ($X_1, X_3 > 0$, $\alpha = 0$). Although analysis on the case where no biofilm is present has been previously done for similar models [96,97], we include it as a baseline for comparing the effect of biofilm on the system—both in the presence and absence of cheaters. For cases (i)-(iii) 2 million simulations were run, however, for case (iv) 500,000 simulations were run. Fewer simulations were used in case (iv) because all simulations resulted in a single outcome—the Tragedy of the Commons, whereas in cases (i)-(iii) we were able to gather information on a variety of outcomes.

Simulations were run in Python 3.9. Each simulation ran for 20,000 time steps to provide ample time for washout to occur and was considered to have achieved stable coexistence if the following two conditions were met: (i) all variants

had a final concentration of at least 0.01 and (ii) each variant made up at least 5% of the total population. These conditions ensure that the total population is sufficiently large and that each variant makes up a sufficiently large portion of the population. This avoids capturing cases where a population collapse or competitive exclusion are occurring, respectively.

The ranges from which parameters were selected varied slightly between the parameters and initial conditions. The cost of quorum sensing ($Q$) and any rate ($D, \beta_X, \beta_E, \alpha$) could not be equal to or greater than 1 to avoid biologically impossible phenomena; for $Q$ this restriction on the range ensures the cost of quorum sensing does not exceed the metabolic intake whereas for the rates it disallows the model from transferring greater than 100% of a population from one space to another since negative population values are biologically irrelevant. The sloughing and adhesion rates ($\beta_X, \beta_E, \alpha$) had a minimum of 0.001, however, we increased the minimum for the cost of quorum sensing and the dilution rate ($Q, D$) to 0.1. The dilution needs to be sufficiently high to ensure washout of non-growing populations and the cost of quorum sensing needs to be high enough to reflect the growth advantage the cheaters have over cooperators; further, increasing the minimum value of the range of these parameters eliminates finding solutions that appear to be stable, but instead have not had sufficient time to washout due to the slow rate and minimal growth difference.

Parameter and initial condition values were randomly selected in each of the 2-million simulations from a log-uniform distribution to examine an equal distribution of different magnitudes with three exceptions; for the dilution ($D$), cost of quorum sensing ($Q$), and the maximum growth rate ($\mu$) a uniform distribution was used.

All the initial conditions ($S(0), E_1(0), X_1(0), X_3(0)$), conversion factors ($\eta, \gamma, \delta$), the nutrient concentration entering the chemostat ($S^0$), and the maximum amount of enzyme and cooperator cells the biofilm can hold ($\hat{E}_2, \hat{X}_2$) were selected from a range of [0.01, 2]. The half-maximal saturation constant of the growth equation ($K_S$) was selected from a range of [0.001, $S^0$] to ensure that the growth of the bacteria was not nutrient limited. Lastly, the maximum growth rate ($\mu$) was selected from a range of [1,100]. The maximum growth rate has the largest range because many different traits can affect a cells growth rate such as metabolism, nutrient uptake rates, and cell size at reproduction, thus, we aimed to leave it less restricted.

All simulations and analysis were completed using Python 3.9. The model was simulated using the solve_ivp function of the SciPy package with the LSODA method of integration. LSODA was chosen because selecting random parameters of varying magnitudes could occasionally cause stiffness. We also implemented two safeguards to account for when solutions tend to 0, causing the machine epsilon to be greater than the solution value potentially creating negative concentrations of the variables. First, we modified the built in tolerances to be smaller by setting the rtol to 2e-10 and the atol to 1e-10. Second, when a variable fell below machine epsilon (2.220446049250313e-16), its value was reset to machine epsilon before calculating the differential equations. NumPy was used to calculate the eigenvalues with the eigvals function from the linalg package. Versions 1.21.4 and 1.10.1 of NumPy and SciPy were used, respectively.

**Isolating the effects of social cheating.** To find a parameter range in which all simulations were stable with only free floating and biofilm cooperators present, we started with the inner quartile range of the stable parameter space. We then adjusted various parameters until 100,000 simulations were all stable, the narrowed ranges are as described in S4 Table; note that initial conditions and parameters not included in S4 Table remained set to their IQR values (S3 Table). Using the same parameter values as the 100,000 stable simulations, we added cheaters to the system by randomly selecting an initial cheater concentration from the full initial condition range described above. Simulations were categorized as follows: stable coexistence occurred when the final concentration of all variants was >= 0.01 and each variant occupied > 5% of the population; competitive exclusion occurred when the conditions for stable coexistence held true for two of the three variants; a Tragedy of the Commons occurred when the final concentration of all variants was <0.01; the remaining simulations were classified as an "other" category and would contain those that had a stable population but where at least one of the variants made up a small proportion of the population.

## Statistical analysis

The Mann-Whitney U Test was performed using SciPy in Python 3.9 to produce boxplot S1 Fig comparing stable parameter and initial condition values. S2 Table shows the reported P-values for parameter or initial condition value compared across all pairwise combinations. The median, mean, and IQR of these stable parameter and initial condition values are reported in S3 Table and was determined by the boxplot function in Python 3.9.

## Supporting information

**S1 Fig. Box and violin plot of stable parameter and initial condition values.** Box plot with the average shown in a dotted line overlays a violin plot of the distribution of stable parameter and initial condition values for the cases where only free floating cooperators are present (light blue), both free floating and biofilm cooperators are present (dark blue), and all variants (both cooperators plus the cheater) are present (red). Parameters or initial conditions randomly selected from a log-uniform distribution have a log Y-axis, and those selected from a uniform distribution have a linear Y-axis. A pairwise two-sided Mann-Whitney U test was for each parameter or initial condition, and the P value is denoted in text above the box/violin plots as *, **, *** for P values $P < .05$, $P < .01$, and $P < .001$ respectively. For explicit P values see S2 Table and for mean and IQR values see S3 Table.
(TIFF)

**S2 Fig. Eigenvalues at varying maximal growth rates.** Scatter plot showing the real (X-axis) and imaginary (Y-axis) parts of the Eigenvalues as the maximum growth rate, $\mu$, increases from 7 to 9 as indicated by the color of the marker. A pair of eigenvalues crosses the imaginary axis where the stable cycles were observed (Fig 4 main text, top), near $\mu = 8.0$, indicating a Hopf bifurcation.
(TIFF)

**S3 Fig. Pairwise scatter plots of all parameter and initial condition values.** Figure components are the same as described in Fig 4 of the main text.
(TIFF)

**S4 Fig. Pairwise scatter plots of parameter values for fully stable outcomes.** Each point represents a single simulation that resulted in full system stability. Graph labels are indicated across the diagonal. Full stable coexistence occurred across a wide range of values.
(TIFF)

**S1 Table. Model parameter and variable descriptions.**
(DOCX)

**S2 Table. *P* values for Mann Whitney U test comparing stable parameter and initial condition values from S1 Fig.**
(DOCX)

**S3 Table. Mean and IQR values from S1 Fig.**
(DOCX)

**S4 Table. Cooperator stable coexistence parameter ranges.**
(DOCX)

**S1 File. Proof the model is well posed.**
(DOCX)

## Author contributions

**Conceptualization:** Bryan K. Lynn, Patrick De Leenheer.

**Formal analysis:** Bryan K. Lynn, Patrick De Leenheer, Benjamin D. Dalziel.

**Methodology:** Bryan K. Lynn, Patrick De Leenheer, Benjamin D. Dalziel.

**Software:** Bryan K. Lynn.

**Supervision:** Patrick De Leenheer, Benjamin D. Dalziel.

**Visualization:** Bryan K. Lynn.

**Writing – original draft:** Bryan K. Lynn, Benjamin D. Dalziel.

**Writing – review & editing:** Bryan K. Lynn, Patrick De Leenheer, Benjamin D. Dalziel.

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
