## [Decision Letter · Decision Letter 0]

22 Sep 2025

PONE-D-25-44591Cooperative Environmental Engineering via Biofilm Formation Can Stabilize Consumer-Resource SystemsPLOS ONE

Dear Dr. Lynn,

Thank you for submitting your manuscript to PLOS ONE. After careful consideration, we feel that it has merit but does not fully meet PLOS ONE’s publication criteria as it currently stands. Therefore, we invite you to submit a revised version of the manuscript that addresses the points raised during the review process.

Please submit your revised manuscript by Nov 06 2025 11:59PM**.** If you will need more time than this to complete your revisions, please reply to this message or contact the journal office at plosone@plos.org. Please include the following items when submitting your revised manuscript:

We look forward to receiving your revised manuscript.

Kind regards,

Bashir Sajo Mienda, PhD

Academic Editor

PLOS ONE

Additional Editor Comments:

Reviewer #1:

Reviewers' comments:

Reviewer's Responses to Questions

**Comments to the Author**

1. Is the manuscript technically sound, and do the data support the conclusions?

Reviewer #1: Yes

2. Has the statistical analysis been performed appropriately and rigorously?

Reviewer #1: I Don't Know

3. Have the authors made all data underlying the findings in their manuscript fully available?

Reviewer #1: Yes

4. Is the manuscript presented in an intelligible fashion and written in standard English?

Reviewer #1: Yes

5. Review Comments to the Author

Reviewer #1: This manuscript addresses the classic cooperation problem in evolutionary and microbial ecology by examining how cooperative behavior can persist in consumer–resource systems despite the threat of social cheating and the tragedy of the commons. The paper’s main contribution is modeling biofilm formation as a form of environmental engineering that helps stabilize cooperation. It also shows that environmental modification is a collective ecological process that is both biologically important and not yet well studied.

The article is well-written, however, I recommend the following revisions:

• The manuscript could more explicitly engage with the literature on partially privatized public goods (e.g., Solutions to the Public Goods Dilemma in Bacterial Biofilms by Drescher et al., 2014; Privatization of Biofilm Matrix in Structurally Heterogeneous Biofilms by Otto et al., 2020) and niche construction theory (e.g., Niche Construction Theory: A Practical Guide for Ecologists by Odling-Smee et al., 2013; An Introduction to Niche Construction Theory by Laland et al., 2016). Without this integration, the manuscript’s novelty may appear overstated.

• In lines 317–320, the authors introduce separate pools for enzymes in the fluid (E1) and biofilm (E2) but doesn’t fully justify this biologically.

• Lines 317–333 describe cooperators, enzymes, and flow processes in detail, much of which is later repeated in Table 1 (lines 363–364). This is similar to lines 345–352, which discuss the assumptions for uptake functions. Present concise definitions in the text, and direct readers to Table 1 for full biological interpretations and units, rather than repeating these twice.

• Lines 270–277 (Discussion): Link your findings to resilience theory, noting that biofilms provide structural feedbacks that increase system resilience by delaying or preventing collapse.

• Lines 283–288 (Discussion): The authors should broaden the interpretation by connecting microbial biofilm resilience to general principles in social-ecological systems (e.g., A General Framework for Analyzing Sustainability of Social-Ecological Systems By Ostrom 2009).

• The authors should conduct a careful proofread for typographical consistency and duplicate references (e.g., line 232 ‘liklihood’ → ‘likelihood’), remove duplicate or redundant references (e.g., line 582 and 588 Sandoz et al. 2007).

6. PLOS authors have the option to publish the peer review history of their article (what does this mean?). If published, this will include your full peer review and any attached files.

Reviewer #1: **Yes: **Musa Hassan Muhammad

---

## [Author Response · Author response to Decision Letter 1]

22 Oct 2025

Thank you for the careful review of our paper. We have attached a file with responses to each point addressed by the editor and reviewer, which is color-coded to separate our responses from their comments. The text from the file is repeated below:

Academic Editor:

Necessary changes to file names and format have been completed to align with the PLOS ONE formatting style.

We used Figshare as a repository, which is listed as an acceptable repository by PLOS ONE. We are unsure of what edits/changes need to be made here.

We have updated our manuscript to be in a LaTeX file format.

We have added some citations that the reviewer recommended or were related to topics the reviewer suggested including.

Reviewer 1:

This manuscript addresses the classic cooperation problem in evolutionary and microbial ecology by examining how cooperative behavior can persist in consumer–resource systems despite the threat of social cheating and the tragedy of the commons. The paper’s main contribution is modeling biofilm formation as a form of environmental engineering that helps stabilize cooperation. It also shows that environmental modification is a collective ecological process that is both biologically important and not yet well studied.

We appreciate your careful review of our manuscript and for the highly constructive comments. We responded to each comment below.

The article is well-written, however, I recommend the following revisions:

• The manuscript could more explicitly engage with the literature on partially privatized public goods (e.g., Solutions to the Public Goods Dilemma in Bacterial Biofilms by Drescher et al., 2014; Privatization of Biofilm Matrix in Structurally Heterogeneous Biofilms by Otto et al., 2020) and niche construction theory (e.g., Niche Construction Theory: A Practical Guide for Ecologists by Odling-Smee et al., 2013; An Introduction to Niche Construction Theory by Laland et al., 2016). Without this integration, the manuscript’s novelty may appear overstated.

Thank you for the suggestion. We agreed that adding more discussion on these topics enhances the quality of the manuscript and arguments made within it. We’ve included additional text in the introduction expanding on partially public goods (lines 79-89) and how environmental engineering and Niche Construction Theory are related (lines 30-38). We also added text into the discussion highlighting how the results of Otto et al. 2020 relate to our results (lines 251-265).

• In lines 317–320 (now 298-299), the authors introduce separate pools for enzymes in the fluid (E1) and biofilm (E2) but doesn’t fully justify this biologically.

We agree that the rationale we included could have been more clear, and we have rewritten our justification (lines 299-301).

• Lines 317–333 (now 297-310), describe cooperators, enzymes, and flow processes in detail, much of which is later repeated in Table 1 (lines 363–364). This is similar to lines 345–352 (now 329-344), which discuss the assumptions for uptake functions. Present concise definitions in the text, and direct readers to Table 1 for full biological interpretations and units, rather than repeating these twice.

Thank you for pointing out the redundant text. The table was merely meant to be an easy reference guide for readers while parsing the text, as such, we have moved the table into supplemental materials (S1 Table) to reduce redundancy.

• Lines 270–277 (now 209-217) (Discussion): Link your findings to resilience theory, noting that biofilms provide structural feedbacks that increase system resilience by delaying or preventing collapse.

We agree that a more explicit connection to resilience theory is warranted. Lines 224-227 have been added to more directly link our results to resilience theory.

• Lines 283–288 (now 222-232) (Discussion): The authors should broaden the interpretation by connecting microbial biofilm resilience to general principles in social-ecological systems (e.g., A General Framework for Analyzing Sustainability of Social-Ecological Systems By Ostrom 2009).

Thank you for this compelling idea to strengthen our discussion. We have added a paragraph linking our system to Social-Ecological System Framework in lines 233-250.

• The authors should conduct a careful proofread for typographical consistency and duplicate references (e.g., line 232 ‘liklihood’ → ‘likelihood’), remove duplicate or redundant references (e.g., line 582 and 588 Sandoz et al. 2007).

Thank you for bringing these errors to our attention. A careful revision of the bibliography and text was done to resolve typographical errors.

---

## [Decision Letter · Decision Letter 1]

16 Nov 2025

Cooperative Environmental Engineering via Biofilm Formation Can Stabilize Consumer-Resource Systems

PONE-D-25-44591R1

Dear Dr. LYNN,

We’re pleased to inform you that your manuscript has been judged scientifically suitable for publication and will be formally accepted for publication once it meets all outstanding technical requirements.

Kind regards,

Bashir Sajo Mienda, PhD

Academic Editor

PLOS ONE

Additional Editor Comments (optional):

Reviewers' comments:

Reviewer's Responses to Questions

**Comments to the Author**

1. If the authors have adequately addressed your comments raised in a previous round of review and you feel that this manuscript is now acceptable for publication, you may indicate that here to bypass the “Comments to the Author” section, enter your conflict of interest statement in the “Confidential to Editor” section, and submit your "Accept" recommendation.

Reviewer #1: All comments have been addressed

2. Is the manuscript technically sound, and do the data support the conclusions?

Reviewer #1: Yes

3. Has the statistical analysis been performed appropriately and rigorously?

Reviewer #1: Yes

4. Have the authors made all data underlying the findings in their manuscript fully available?

Reviewer #1: (No Response)

5. Is the manuscript presented in an intelligible fashion and written in standard English?

Reviewer #1: Yes

6. Review Comments to the Author

Reviewer #1: (No Response)

7. PLOS authors have the option to publish the peer review history of their article (what does this mean?). If published, this will include your full peer review and any attached files.

Reviewer #1: **Yes: **Musa Hassan Muhammad

---

## [Editor Report · Acceptance letter]

PONE-D-25-44591R1

PLOS ONE

Dear Dr. Lynn,

I'm pleased to inform you that your manuscript has been deemed suitable for publication in PLOS ONE. Congratulations! Your manuscript is now being handed over to our production team.

Kind regards,

on behalf of

Dr. Bashir Sajo Mienda

Academic Editor

PLOS ONE